# A Cross-Sectional Study of Oral Health Status and Behavioral Risk Indicators among Non-Smoking and Currently Smoking Lithuanian Adolescents

**DOI:** 10.3390/ijerph20166609

**Published:** 2023-08-19

**Authors:** Sandra Petrauskienė, Miglė Žemaitienė, Eglė Aida Bendoraitienė, Kristina Saldūnaitė-Mikučionienė, Ingrida Vasiliauskienė, Jūratė Zūbienė, Vilija Andruškevičienė, Eglė Slabšinskienė

**Affiliations:** Department of Preventive and Paediatric Dentistry, Faculty of Dentistry, Academy of Medicine, Lithuanian University of Health Sciences, Lukšos-Daumanto 6, LT-50106 Kaunas, Lithuania; migle.zemaitiene@lsmu.lt (M.Ž.); egleaida.bendoraitiene@lsmu.lt (E.A.B.); kristina.saldunaite@lsmu.lt (K.S.-M.); ingrida.vasiliauskiene@lsmu.lt (I.V.); jurate.zubiene@lsmu.lt (J.Z.); vilija.andruskeviciene@lsmu.lt (V.A.); egle.slabsinskiene@lsmu.lt (E.S.)

**Keywords:** adolescent, dental caries, periodontal status, smoking, oral health

## Abstract

The purpose of this study was to evaluate oral health status, behavioral risk indicators, and the impact of smoking on oral health among Lithuanian adolescents. This representative cross-sectional study was conducted among 15-year-old Lithuanian adolescents. The method of multistage cluster sampling was used. A total of 1127 adolescents met the inclusion criteria. Two originally created self-reported questionnaires were used in this study. Dental caries, periodontal status, and oral hygiene status were evaluated by four trained researchers. A *p*-value ≤ 0.05 was set to indicate statistically significant differences. Statistical analysis included Mann–Whitney, Kruskal–Wallis, and Spearman correlation tests. Out of all the participants, 9.6% self-reported being a current tobacco smoker. The mean PI value was 1.14 ± 0.69 among all the participants. Currently smoking adolescents had more active caries lesions (D-S) than those who did not smoke (13.2 ± 16.4 vs. 9.8 ± 10.7, *p* = 0.023). Considering periodontal status, non-smoking adolescents had significantly lower mean PSR index scores than current smokers (0.52 ± 0.51 vs. 0.61 ± 0.50, *p* = 0.0298). Tobacco smoking and the consumption of energy drinks were significantly associated (OR = 3.74, 95% CI 2.66–5.26, *p* < 0.001) among participants. Currently smoking adolescents tended to have improper dietary habits, especially a higher consumption of energy drinks; thus, they were more likely to have active dental caries lesions, as well as poorer periodontal status, than their non-smoking peers.

## 1. Introduction

Across the world, dental caries and periodontal diseases are prevalent, leading to the loss of teeth throughout the life course of individuals [1]. However, differences in pathophysiology and nutritional imbalances in microbiota both affect the development of dental caries and periodontal diseases [2]. Recently, dental caries has been highlighted as a non-communicable disease strongly influenced by personal behaviors and lifestyles [3]. Because biofilm dysbiosis is known to lead to initiation and progression of active dental caries [3], proper attention should be given to the early stages of this process, such as non-cavitated caries lesions, to prevent the further development of disease [4]. It has been proven that irreversible periodontal tissue changes may start in adolescence [5].

Health behavior first develops in childhood and plays an important role in determining health later in life [6]. As a period of life, adolescence is associated with increased independence from parents and changes in behavior with respect to lifestyle and diet [7]. Attitudes toward health formed during adolescence may persist into adulthood [8]. However, adolescents today face many health and social challenges; these include major biological, psychosocial, and cognitive transitions [9]. Often, adolescents consume an insufficient amount of vegetables and fruits [10]. During adolescence, unproper dietary habits are related to risk behavior, and they should be improved to prevent health impairment [11]. During this period, personal oral hygiene tends to be less of a priority due to increased independence [12]. As a result, unhealthy habits adopted during adolescence may lead to adverse oral health outcomes [13,14]. The WHO strongly recommends that human consumption of free sugars should represent less than 10% of total energy intake [15]. However, adolescents tend not to follow international dietary recommendations [8], especially due to misleading marketing campaigns of sports and energy drinks [16]. An increased intake of sugar-containing foods and beverages such as energy drinks with no nutritional value is therefore considered a risk factor for dental caries, having consequent effects upon dentition [16,17,18].

Worldwide, 22.3% of the population uses tobacco products [19]. In Lithuania, 18.9% of adult citizens describe themselves as smokers [20]. According to the WHO, a current smoker is an individual aged 15 years or older who uses tobacco products on a daily or non-daily basis [21]. Smoking as a habit is related to various factors, such as parental education, gender, and behavior patterns [22]. Tobacco usage has a direct effect on general health; therefore, unhealthy behavior patterns formed in adolescence can be the leading cause of severe diseases in adulthood [23]. Smoking impairs the functions of the respiratory and nervous systems. It also has an effect on depression and manifestation of psycho-emotional fatigue symptoms [24,25]. Nicotine can have a negative impact on memory, attention and concentration, academic achievements, and cognition [25].

Smoking has a detrimental impact not only on health in general, but also on oral health status in particular [1,26]. A smoking habit affects the risk of dental caries and periodontal disease because smokers are more likely to have poor oral health behavior and are less likely to visit a dentist regularly [27,28,29]. Thus, smoking adolescents tend to have improper oral hygiene habits, poor oral hygiene status, and an increased risk of dental caries [30,31]. Consequently, a deterioration in periodontal health as a result of smoking may begin even in adolescence [32,33], especially when associated with a high intake of sugary beverages [34]. Researchers have now proved an association between smoking and a number of unhealthy diet behaviors, including a more frequent choice of junk foods, a higher consumption of energy drinks, and irregular breakfast patterns [35,36,37,38].

In Lithuania, the prevalence of smoking among adolescents has been determined to be between 12.9% and 16.5% [20,39]. Among Lithuanian adolescents, dental caries is highly prevalent. One recent report indicated that 75.1% of 15-year-olds in Lithuania have dental caries [40]. However, research into associations between oral health and oral-health-related behavioral indicators such as dietary habits, dental attendance, and current levels of tobacco smoking among adolescents has been scarce in Lithuania. We hypothesized that currently smoking adolescents will have more active caries lesions and worse periodontal tissue status than non-smoking participants. In this study, therefore, we sought to investigate how tobacco smoking is related to oral health and other oral-health-related behavioral indicators among Lithuanian adolescents.

## 2. Materials and Methods

### 2.1. Study Population

This cross-sectional study was a large population-based research project that considered the epidemiology of dental caries, as well as periodontal and oral hygiene status, among older school-age children in Lithuania during the 2013/2014 academic year.

This study was performed in the public schools located in the centers of 10 Lithuanian counties. A multistage cluster sampling method was used to define a representative sample. Each county was divided into smaller administrative units, and schools were picked out from the alphabetic list of all schools, provided by Centre of Information Technologies in Education (the first and the last schools from the list were chosen). A total of 2000 adolescents from 20 schools all over the country were approached. The inclusion criteria for participants were as follows: an age of 14.5–15.5 years; completion of the study questionnaire; and agreement to be enrolled into the study by means of written informed consent of both parents and participant as well. A total of 1127 adolescents met the inclusion criteria. The study was voluntary; participants were informed that they could withdraw from the study at any time. The flow chart of the study sample is presented in Figure 1.

### 2.2. Sample Size Calculation

The sample size was calculated using Paniott’s formula with an error of 0.05%, based on the number of 15-year-olds in the Lithuanian population in 2012, which was 33,163 according to Statistics Lithuania. Using Paniott’s formula, it was determined that no less than 396 of 15-year-olds had to be included in the study. 

### 2.3. Ethical Approval

Permission to examine the subjects was granted by the Kaunas Regional Biomedical Research Ethics Committee (dated 27 November 2012; No. BE-2-47). The aims and procedures of the study were explained to the parents of study participants, and written informed consent was obtained from each adolescent who took part.

### 2.4. Instruments and Clinical Examination

The pilot study was carried out in the one selected school (‘Saule’ gymnasium in Kaunas City) in January of 2013. The pilot study enrolled thirty-five 15-year-old subjects, who were not included in the final sample.

The Cronbach‘s alpha coefficient was calculated to evaluate the internal consistency of the questionnaire and was found to be 0.8 (a good value). Questionnaires’ validation was performed through the evaluation of content validity. Finally, the validation of the questionnaire was performed regarding the three experts, given recommendations. After 7 days, questionnaires were distributed repeatedly for the same respondents to complete. The results of test–retest reliability were equal to 0.77 (0.75–0.8). The final versions of questionnaires were created after the analysis of the given comments of both participants and their parents during the pilot study.

Two originally created self-reported questionnaires were used in this study: one for adolescent participants and another for their parents. The adolescent questionnaire consisted of twenty questions assessing demographic characteristics (adolescent’s age, gender, grade, and living area), behavioral habits such as oral hygiene, measures used for oral hygiene, dental attendance, cigarette smoking, personal nutrition, and self-reported oral health status. The questionnaire for parents included thirty-eight items, but only one of these, concerning parental (maternal and paternal) education, was assessed for the purposes of the study. Fathers and mothers provided their answers separately.

This self-administered questionnaire included two questions related to smoking habit. At the beginning, all the adolescents were asked if they had ever smoked. A smoking habit was assessed with regard to the frequency of smoking at the time of the study. Participants could say that they smoked ‘daily’; ‘at least once a week, but not daily’; ‘less often than once a week’; or that they ‘do not smoke’. Later, all responses were dichotomized as indicating either a non-smoker (‘do not smoke’) or a current smoker (‘daily’; ‘at least once a week, but not daily’; or ‘less often than once a week’).

Dental attendance patterns were measured by asking each participant whether they had had a routine dental check-up during the last 12 months; the response alternatives were simply ‘yes’ or ‘no’.

Dietary behavior was analyzed in terms of the frequency with which participants consumed fruits, vegetables, sweets, and energy drinks. The response alternatives were ‘never’, ‘less than once a week’, ‘once a week’, ‘2–4 days a week’, ‘5–6 days a week’, ‘once a day’, or ‘more than once a day’. For the purposes of analysis, dietary behavior with respect to fruits, vegetables, and sweets was dichotomized into two groups: low consumption (‘never’, ‘less than once a week’, or ‘once a week’) and high consumption (‘2–4 days a week’, ‘5–6 days a week’, ‘once a day’, or ‘more than once a day’). With regard to intake of energy drinks, participants were similarly classified into those who did not consume (‘never’) and those who did consume (‘less than once a week’, ‘once a week’, ‘2–4 days a week’, ‘5–6 days a week’, ‘once a day’, or ‘more than once a day’).

The education level of participants’ parents was classified as follows: low education (did not finish secondary school); moderate education (graduated from secondary school); and high education (graduated from higher education or university).

Frequency of toothbrushing was classified into three groups as follows: twice a day; once a day; and irregular toothbrushing (‘at least once a week, but not daily’; ‘less often than once a week’; or ‘never’).

Clinical examinations were performed under standardized conditions using a comfortable chair with a head support and portable dental units equipped with a halogen light source, compressed air, and suction device. All oral status measurements were performed by four researchers (J.Z., K.S.-M., M.Ž., and V.A.) who were fully trained. Training was also given to thirty-five 15-year-old subjects who were not included in the final sample. The kappa value for inter-examiner reliability was 0.92; for intra-examiner reliability, the values ranged from 0.92 to 0.94.

Caries experience was measured with the component D-S of the decayed, missing, and filled surfaces (DMF-S) index. Dental caries (D-S) was recorded using clinical criteria based on the assessment of lesion activity [41]. Surfaces that were active (intact lesion), active (discontinuity), or active (cavity), as well as fillings with active caries, were classified as active. All other surfaces, i.e., those which were inactive (intact lesion), inactive (discontinuity), or inactive (cavity), as well as fillings with inactive caries, were classified as inactive [41]. Considering the severity of caries, active lesions were dichotomized as either non-cavitated (intact lesion) or cavitated (discontinuity, cavity, and filling with active caries) [41,42]. Surfaces of teeth were classified as either smooth (buccal and lingual), proximal (mesial and distal), or occlusal.

Periodontal health was evaluated using the Periodontal Screening and Recording (PSR) index [43]. The dentition was divided into sextants, each tooth was probed, and the highest code (0 to 4) of each sextant was recorded. The highest PSR index code of all sextants was then used as a final score for diagnosis [43]. Finally, oral hygiene status was assessed using the Silness–Löe plaque index (PI) with ratings as follows: 0—excellent; 0.1–0.9—good; 1.0–1.9—fair; and 2.0–3.0—poor [43].

### 2.5. Statistical Analysis

Statistical data analysis was conducted using SPSS 27.0 software (Chicago, IL, USA). The Kolmogorov–Smirnov test was employed for the investigation of hypotheses about the normality of parameter distribution. Continuous data (PSR index, Silness–Löe plaque index, and active caries D-S) of two independent samples were compared using the Mann–Whitney U test. The DMFT, PSR, and PI indices were represented as the mean ±  SD. The independence of categorical data was evaluated with the help of the chi-squared (*χ*^2^) test. Binary logistic regression analysis was performed for risk prediction. Two independent variables based on comparative analysis were entered into the model of multivariate logistic regression: current smoking, and poor oral hygiene. Correlation between oral hygiene status and active dental caries lesions was determined using Spearman’s correlation coefficient (rho).

The difference was considered to be statistically significant when the *p*-value was less than 0.05. 

## 3. Results

Among all study participants, 44.9% reported that they had smoked at least once, and 9.6% self-identified as a current tobacco smoker (Table 1). With respect to gender, girls constituted the majority (59.6%) of the total study population (Table 1). Results also showed that the majority (66.5%) of participants lived in urban areas (Table 1). Girls were more likely to be current tobacco smokers than boys (51.0% vs. 49.0%, *p* > 0.05). The proportion of currently smoking adolescents was higher in urban areas than rural areas (68.3% vs. 31.7%, *p* > 0.05). Considering parental education, a high level of education prevailed among both mothers (61.1%) and fathers (49.5%). Binary logistic regression analysis showed the likelihood of being a current smoker was 2.75 times greater (95% CI 1.64–4.60, *p* < 0.001) for an adolescent who had a mother with low education; if a father had low education, the corresponding figure was 2.70 (95% CI 1.55–4.70, *p* < 0.001) (Table 2).

Overall, 60.7% of participants reported brushing their teeth twice a day (Table 3). Results of the clinical examinations revealed that the mean PI score was 1.14 ± 0.69 among all participants (Table 4). Evaluation of periodontal status showed that the mean PSR index score was 0.53 ± 0.51 among all participants; however, non-smoking adolescents had significantly lower mean PSR scores than current smokers (0.52 ± 0.51 vs. 0.61 ± 0.50, *p* = 0.0298) (Table 4). Binary logistic regression analysis revealed a greater likelihood of affected periodontal tissues (PSR ≥ 1) among adolescents who smoked, as well as among those with fair or poor oral hygiene status (Table 2).

Overall, the mean D-S score of active caries lesions (both non-cavitated and cavitated) was 10.18 ± 11.33 among participants. Adolescents who smoked had significantly higher active mean D-S scores than those who did not smoke (13.2 ± 16.4 vs. 9.8 ± 10.7, *p* = 0.023). Considering tooth surfaces, significantly lower active mean D-S scores were recorded for the smooth and proximal surfaces of non-smokers, compared with current smokers (*p* < 0.05) (Table 4). In addition, currently smoking participants had significantly higher active non-cavitated mean D-S scores than those who did not smoke (12.12 ± 15.15 vs. 8.91 ± 10.07, *p* = 0.037) (Table 4). With respect to gender, non-smoking girls had statistically significantly lower mean D-S scores for smooth surface, proximal surface, and non-cavitated active caries lesions than non-smoking boys; these scores were 4.37 ± 4.49 vs. 6.27 ± 6.38, *p* < 0.001; 3.26 ± 3.94 vs. 5.06 ± 6.22, *p* < 0.001; and 7.40 ± 7.97 vs. 11.21 ± 12.32, *p* < 0.001, respectively (Table 5). In addition, significantly higher active mean D-S scores for proximal surface and non-cavitated lesions were recorded for currently smoking girls, compared with non-smoking girls; these scores were 5.04 ± 3.94 vs. 3.26 ± 3.94, *p* = 0.002, and 10.11 ± 7.80 vs. 7.40 ± 7.97, *p* = 0.017), respectively (Table 5).

Table 6 presents values for Spearman’s correlation coefficient between mean PI and active D-S scores in both groups: non-smokers and currently smoking adolescents. Oral hygiene status had low (poor) significant positive correlations with active mean D-S scores of proximal, smooth surfaces, non-cavitated, and cavitated caries lesions among non-smoking participants. Meanwhile, no significant correlation was observed between mean PI and active D-S scores among currently smoking adolescents (Table 6).

Various factors affecting the oral health of participants, such as dental attendance and dietary habits, were also evaluated. The results showed that statistically significantly more non-smoking adolescents (65.0%) reported regular visits to oral health specialists than those who currently smoked (52.9%) (*p* = 0.014) (Table 3). Significantly more currently smoking adolescents reported that they consumed energy drinks, compared with their non-smoking peers (62.5% vs. 31.2%, *p* < 0.001), and this association was significant (OR = 3.74, 95% CI 2.66–5.26, *p* < 0.001). Results also showed that more currently smoking adolescents (81.4%) reported frequently consuming sweets than non-smokers (76.0%) (*p* = 0.226) (Table 3). Finally, more non-smokers reported a high intake of vegetables and fruits than currently smoking participants (90.8% vs. 80.6%, *p* < 0.001).

## 4. Discussion

In this study, we sought to investigate oral health status and behavioral risk indicators among non-smoking and currently smoking Lithuanian adolescents. Study results revealed that almost half of 15-year-old Lithuanian adolescents had smoked at least once. However, a tenth of participants self-reported as current tobacco smokers. Currently smoking adolescents had significantly higher mean PSR index scores, indicating poorer periodontal health, and higher mean values for active dental caries (D-S), compared with their non-smoking counterparts. In addition, currently smoking adolescents were less likely to make regular dental visits and more likely to have improper dietary habits than their non-smoking peers. A significantly higher prevalence of energy drink consumption was also recorded among currently smoking participants. Considering dietary habits along with the clinical oral findings, we found that smoking played a role as a negative oral-health-behavior-related indicator. Such young people can be seen as constituting a population of patients that requires exceptional attention by oral health specialists, in order to reinforce smoking prevention measures. Efforts need to be focused on motivating adolescents to make routine dental visits, to build and maintain healthy eating habits, and to break unhealthy habits. Oral health specialists who encourage their patients to cease smoking may help them to reduce their risk of developing not only oral diseases, but also general diseases.

This study revealed that 9.6% of 15-year-old Lithuanian adolescents were current smokers. In other countries, the prevalence of smoking among teenagers varies from 6.3% to 17.2% [44,45,46]. Education can be considered a strong predictor of smoking [47]. Reports in the literature have indicated that regular smoking is less common among adults with higher levels of education [47]. Among currently smoking adolescents, maternal education tends to play a more important role than paternal education [44,48]. The study carried out by Staff et al. showed that teenagers were more likely to smoke if their mothers had low education [49]. However, another study conducted in Mexico revealed a contrary finding: current smoking among adolescents was related to having a mother with better education who also smoked [44]. In Turkey, higher maternal education was found to be related to smoking among female adolescents, while lower maternal education was a predictor for smoking among adolescent males [48]. Our findings revealed that adolescents who had a mother or father with low education were significantly more likely to smoke. This conclusion is supported by the findings of a study carried out in Indonesia [50]. Contrarily, Barreto et al. reported that smoking among high school students was not associated with maternal education [45].

In the present study of Lithuanian adolescents, an overwhelming majority (89.9%) of participants reported high consumption of fruits and vegetables; in contrast, studies of Portuguese and Italian schoolchildren found a less frequent consumption of these products [51,52]. However, we did find that high consumption of fruits and vegetables was significantly less common among currently smoking Lithuanian adolescents than among non-smokers, and this finding is in line with the results of other studies [53,54]. In the present study, we also found that 80% of Lithuanian adolescents reported a high consumption of sweets. In other countries, adolescents have been found to consume sweets considerably less frequently [51,52]. A study carried out in Germany confirmed that the intake of beverages was significantly associated with DMF-S score among 15-year-olds [55]. With respect to oral health, these are noteworthy findings because high consumption of added sugar may lead to higher prevalence of periodontal diseases and increase the risk of dental caries [1,56].

Energy drinks are popular worldwide, despite reported detrimental effects upon general and oral health [57]. One major concern is that consumption of energy drinks is positively related not only to regular smoking, but also to increased risk-taking behavior among adolescents [58,59]. Considering oral health, the smoking adolescents consuming energy drinks tended not to follow recommended toothbrushing twice a day [59]. Since 2014, the sale of energy drinks to under-18-year-olds has been banned by law in Lithuania [60]. As of yet, there are no data to confirm whether this regulation has had any impact on energy drink consumption among Lithuanian adolescents. In the present study, we found an association between consumption of energy drinks and the presence of a smoking habit among adolescent participants; another study carried out in Italy came to a similar conclusion [61].

This study revealed that twice-a-day toothbrushing prevailed among participants. We also found that a lower proportion of current smokers brushed their teeth as recommended than non-smokers; however, this result was statistically insignificant. In contrast, a study carried out in Finland showed that a majority of smoking adolescents brushed their teeth only once daily [36,62].

Considering studies published across the world, it is probably the case that, during adolescence, the experience and prevalence of dental caries is more of an actual issue than periodontal diseases. Nevertheless, available scientific evidence clearly demonstrates the detrimental effects of smoking on oral health and on the development of both caries and periodontal diseases [26,63], and this was confirmed in the present study. Previous studies have revealed associations between smoking and caries development in adolescents and young adults [27,31,64]. In addition, a study carried out in Finland revealed that poor oral health behavior among young smokers leads to an increased demand for restorative dental treatment later in life [28,29]. In the current study, we focused on the examination of active caries lesions because of the possibility that the process of their development can be arrested if oral health behavior and oral hygiene skills are improved [65].

The findings of this study also revealed that non-smoking adolescents had better periodontal status than their smoking peers. Another study performed in Lithuania showed a similar trend in 18-year-olds without regard to smoking status [66]. In our study, both factors, i.e., smoking and fair/poor oral hygiene, were found to have a detrimental impact on periodontal health status; this was in line with the results of other studies [26,32,63]. Studies carried out in Finland and Greece found that smoking adolescents exhibited a high burden of gingival bleeding [29,67]; in contrast, another study carried out in Nigeria did not reveal any significant association in this regard [63]. However, different indices of periodontal examination such as CPI and BOP were employed in these reported studies; thus, any comparison of periodontal health status might be misleading.

This representative study was the first to assess and evaluate the relationship between oral health behavior, dietary and smoking habits, and the experience of active (non-cavitated and cavitated) dental caries and periodontal health status among 15-year-old Lithuanian adolescents.

Some limitations of the present study should be acknowledged. This study was based on self-reported questionnaires. Both questionnaires consisted of closed-ended questions, which may have limited and narrowed the relevance of the responses. The data related to smoking, dietary, and oral hygiene habits were self-reported and subjective. The study did not cover information regarding to the oral health behaviors of parents, especially tobacco smoking, which may have had an impact on their child habits. Finally, the oral examinations were based on clinical evaluation without radiography; because of this, the prevalence of dental caries and poor periodontal status may have been underestimated.

## 5. Conclusions

In this study, we found that adolescent smokers were more likely to make irregular dental visits and have improper dietary habits, especially the consumption of energy drinks; thus, they had more active dental caries lesions and poorer periodontal status than their non-smoking peers. Adolescent smokers also faced a higher risk of developing oral diseases than non-smokers. 

## Figures and Tables

**Figure 1 ijerph-20-06609-f001:**
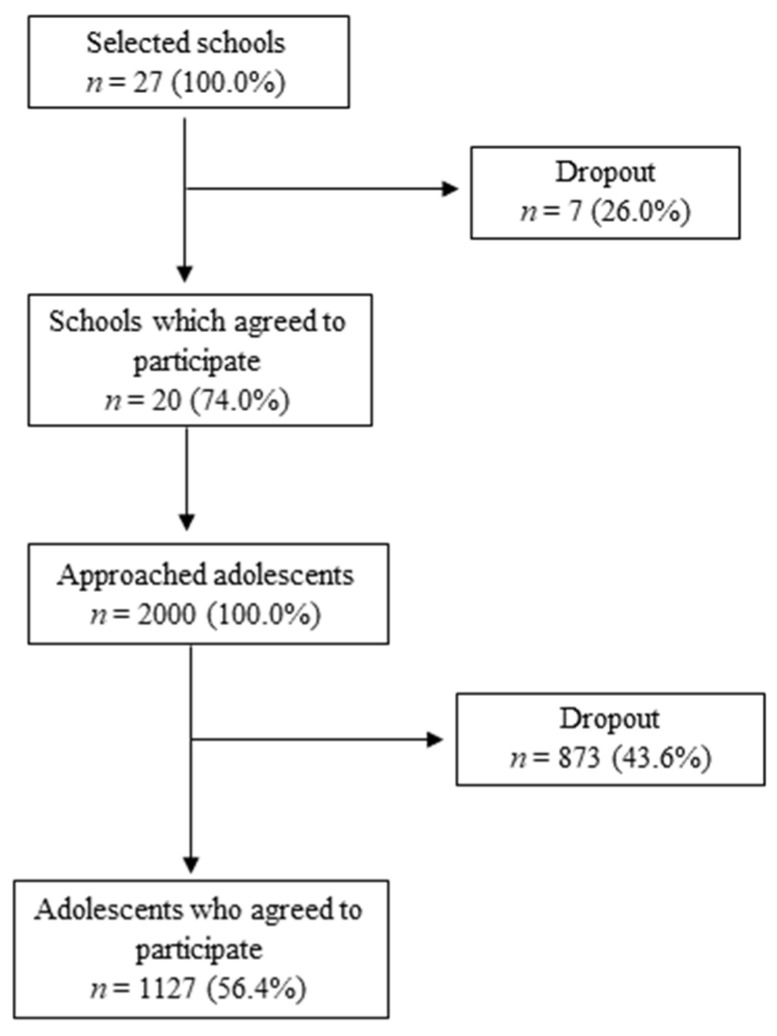
Flowchart diagram of study sample formation following multistage cluster sampling.

**Table 1 ijerph-20-06609-t001:** Demographic characteristics of participants by smoking status (*n* = 1127).

Variables	Smoking Status		Total *n* (%)	*p*-Value
	Non-Smoker *n* (%)	Current Tobacco Smoker *n* (%)		
Gender (missing *n* = 51)				
Boy	385 (39.5)	51 (49.0)	436 (40.4)	0.06
Girl	589 (60.5)	53 (51.0)	642 (59.6)	
Total	974 (100.0)	104 (100.0)	1078 (100.0)	
Living area (missing *n* = 51)				
Urban	646 (66.3)	71 (68.3)	717 (66.5)	0.69
Rural	328 (33.7)	33 (31.7)	361 (33.5)	
Total	974 (100.0)	104 (100.0)	1078 (100.0)	
Maternal education (missing *n* = 147)				
Low	91 (10.3)	23 (24.0)	114 (11.6)	<0.001 ^a^
Moderate	239 (27.0) ^a^	28 (29.2) ^a^	267 (27.3)	
High	554 (62.7) ^a^	45 (46.8) ^a^	599 (61.1)	
Total	884 (100.0)	96 (100.0)	980 (100.0)	
Paternal education (missing *n* = 309)				
Low	75 (10.2)	20 (23.5)	95 (11.6)	<0.001 ^a^
Moderate	282 (38.5) ^a^	36 (42.4) ^a^	318 (38.9)	
High	376 (51.3) ^a^	29 (34.1) ^a^	405 (49.5)	
Total	733 (100.0)	85 (100.0)	818 (100.0)	

Chi-squared test, comparing results by smoking status (non-smoker or current tobacco smoker). ^a^ significant difference between smokers and non-smokers.

**Table 2 ijerph-20-06609-t002:** Binary logistic regression model results.

Characteristics	Category	OR (95% CI)	*p*
PSR ≥ 1			
Smoking	Current tobacco smoker	1.77 (1.03–3.04)	0.039
Oral hygiene status	PI ≥ 1	0.039	<0.001
Current tobacco smoker			
Maternal education	Low	2.75 (1.64–4.60)	<0.001
Paternal education	Low	2.70 (1.55–4.70)	<0.001

OR—odds ratio, CI—confidence interval.

**Table 3 ijerph-20-06609-t003:** Adolescents’ (*n* = 1127) oral hygiene status, dental attendance, and consumption of various products by smoking status.

Variables	Smoking Status		Total *n* (%)	*p*-Value
	Non-Smoker *n* (%)	Current Tobacco Smoker *n* (%)		
Frequency of toothbrushing (missing *n* = 53)				
Twice a day	597 (61.6)	55 (52.9)	652 (60.7)	0.103
Once a day	333 (34.3)	41 (39.4)	374 (34.8)	
Irregularly	40 (4.1)	8 (7.7)	48 (4.5)	
Total	970 (100.0)	104 (100.0)	1074 (100.0)	
Regular dental visits (missing *n* = 51)				
Yes	632 (65.0) ^a^	54 (52.9) ^a^	686 (63.8)	0.014 ^a^
No	341 (35.0)	49 (47.1)	390 (36.2)	
Total	973 (100.0)	103 (100.0)	1076 (100.0)	
Consumption of fruits/vegetables (missing *n* = 58)				
Low	89 (9.2)	19 (19.4)	108 (10.1)	0.001 ^a^
High	882 (90.8) ^a^	79 (80.6) ^a^	961 (89.9)	
Total	971 (100.0)	98 (100.0)	1069 (100.0)	
Consumption of sweets (missing *n* = 60)				
Low	233 (24.0)	18 (18.6)	251 (23.5)	0.226
High	737 (76.0)	79 (81.4)	816 (76.5)	
Total	970 (100.0)	97 (100.0)	1067 (100.0)	
Consumption of energy drinks (missing *n* = 224)				
No	550 (68.8)	39 (37.5)	589 (65.2)	<0.001 ^a^
Yes	249(31.2) ^a^	65 (62.5) ^a^	314 (34.8)	
Total	799 (100.0)	104 (100.0)	903 (100.0)	

Chi-squared test, comparing results by smoking status (non-smoker and current tobacco smoker). ^a^ significant difference between smokers and non-smokers.

**Table 4 ijerph-20-06609-t004:** Relationship between active lesions, oral hygiene status, periodontal status, and current smoking status among participants (*n* = 1127).

Variables	Smoking Status		Total MS ± SD	*p*-Value
	Non-Smoker MS ± SD	Current Tobacco Smoker MS ± SD		
Active caries of surfaces (D-S) (missing *n* = 0)				
Smooth	5.12 ± 5.34	6.75 ± 7.76	5.35 ±5.68	0.039
Proximal	3.97 ± 5.04	5.66 ± 7.87	4.14 ± 5.36	0.034
Occlusal	0.68 ± 1.22	0.79 ± 1.43	0.7 ± 1.24	0.464
Active caries on cavitation (D-S) (missing *n* = 0)				
Non-cavitated	8.91 ± 10.07	12.12 ± 15.15	9.30 ± 10.64	0.037
Cavitated	0.00 ± 0.00	1.00 ± 0.00	0.89 ± 1.89	0.314
Oral hygiene status (missing *n* = 11)				
PI	1.13 ± 0.70	1.21 ± 0.65	1.14 ± 0.69	0.152
Periodontal status (missing *n* = 0)				
PSR index	0.52 ± 0.51	0.61 ± 0.50	0.53 ± 0.51	0.0298

MS ± SD—mean score and standard deviation; statistical analysis by Mann–Whitney U test.

**Table 5 ijerph-20-06609-t005:** Relationship between active caries, gender, and current smoking status among participants (*n* = 1127).

Variables	Gender	Smoking Status		*p*-Value
		Non-Smoker MS ± SD	Current Tobacco Smoker MS ± SD	
Active caries of surfaces (D-S)				
Smooth	Boy	6.27 ± 6.38	8.02 ± 10.16	0.091
	Girl	4.37 ± 4.49	5.5 ± 4.12	0.069
	*p*-value	<0.001 ^a^	0.878	
Proximal	Boy	5.06 ± 6.22	6.31 ± 10.51	0.220
	Girl	3.26 ± 3.94	5.04 ± 3.94	0.002 ^b^
	*p*-value	<0.001 ^a^	0.065	
Occlusal	Boy	0.78 ± 1.40	0.98 ± 1.76	0.362
	Girl	0.61 ± 1.09	0.60 ± 0.99	0.944
	*p*-value	0.152	0.0550	
Active caries on cavitation (D-S)				
Non-cavitated	Boy	11.21 ± 12.32	14.20 ± 20.02	0.137
	Girl	7.40 ± 7.97	10.11 ± 7.80	0.017 ^b^
	*p*-value	<0.001 ^a^	0.616	
Cavitated	Boy	0.90 ± 1.74	1.12 ± 2.41	0.434
	Girl	0.84 ± 1.96	1.06 ± 1.95	0.430
	*p*-value	0.136	0.487	

MS ± SD—mean score and standard deviation; statistical analysis by Mann–Whitney U test. ^a^ significant difference between genders. ^b^ significant difference between smokers and non-smokers.

**Table 6 ijerph-20-06609-t006:** Correlation between oral hygiene status (PI) and active D-S scores in both groups (smoking status).

Oral Hygiene Status	Smoking Status		Active DS Of Surfaces			Cavitation of Active DS	
			Smooth	Proximal	Occlusal	Non-Cavitated	Cavitated
PI	Non-smoker	rho	0.234 **	0.216 **	0.150 **	0.217 **	0.238 **
		*p*-value	<0.001	<0.001	<0.001	<0.001	<0.001
	Current tobacco smoker	rho	0.059	−0.037	0.046	0.020	0.012
		*p*-value	0.555	0.711	0.639	0.838	0.901

Rho—Spearman ranking correlation coefficient. ** Correlation is significant at the 0.01 level (2-tailed). Variables statistically associated (*p* < 0.01).

## Data Availability

The datasets used and/or analyzed during the current study are available from the corresponding author on reasonable request.

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
