# Peer review of "A Cross-Sectional Study of Oral Health Status and Behavioral Risk Indicators among Non-Smoking and Currently Smoking Lithuanian Adolescents"

_ijerph, 2023, doi:10.3390/ijerph20166609_

Round 1

Reviewer 1 Report

Dear Authors,

I have completed my evaluation of the article. I appreciate the authors’ efforts in preparing the article. Generally, it is clear that the oral status of smokers or alcohol consumers is poor.

The novelty of the article is low. The authors assessed the oral hygiene of non-smokers and smokers adolescents in Lithuania. Smoking is highlighted as a predisposing factor for increasing caries and periodontal diseases. The article provides no new information about non-smokers and smokers. The conclusion is brief and reports the most important findings of the research.  

However, the vast number of participants (i.e., 1127) is one distinguishing feature of this present study. Here is my comment:

Please revise as follows: The DMFT, PSR, and PI indices were represented as the mean ± SD.

It would be better if the authors, while examining the participants, reported on another disease (e.g., nicotinic stomatitis and frictional hyperkeratosis).

Author Response

Our responses to the Reviewer‘s comments

We thank the reviewers for their constructive comments. Please find enclosed our responses to the reviewer‘s highlighted in red.

  1. Please revise as follows: The DMFT, PSR, and PI indices were represented as the mean ± SD. Amended as suggested.

  1. It would be better if the authors, while examining the participants, reported on another disease (e.g., nicotinic stomatitis and frictional hyperkeratosis). This study design included DMFT/DMFS, PI and PSR indices, thus another diseases like nicotinic stomatitis and frictional hyperkeratosis were not examined.

Reviewer 2 Report

A Cross-Sectional Study of Oral Health Status and Behavioral Risk Indicators among Non-Smoking and Currently Smoking Lithuanian Adolescents

-       -   Authors should cut the word “to date” in sentence line no.80, page no. 2 because the research ethics committee of this study has been done since 2012.

-     -     Authors should explain the detail about a specific study conducted time and the duration study period.

-     -     Authors should declare how to calibrate the reliability and internal consistency of the questionnaires.

-    -      How many items are on the questionnaire of adolescent participants? Please specify.

-    -      Authors should discuss demographic characteristics of participants not only maternal and parental education but should also discuss gender and living area.

-     -     The one of important factors is socio-economic status but it was not included in this study. Please describe.

-      -    Authors should discuss about the frequency of toothbrushing and consumption of sweets and why were not significant among smoking habits. These factors are important to dental caries and periodontal disease. Authors should discuss these issues.

-    -      Authors should give the details of smoking habit such as the type of cigarette, frequency of smoking per day, and how long the daily smoking habit.

-          Authors should explain why the consumption of fruits and vegetables and energy drinks is related to dental caries and periodontal disease.

-  -        Authors should explain why the results of PI were not a significant difference between smoking habit groups but the PSR index showed a significant difference. It might be the other factors that cause periodontal disease.

-     -     Only active caries on proximal has a significant difference between smoking habit groups. So the conclusion from this study might not conclude the smoking group has more tendency to dental caries and periodontal disease than the non-smoking group.

-    -      The references need to be replaced with recent references. Approximately 40% of the references should be from the last 2-3 years and 90% from the last 10 years.

Author Response

Our responses to the Reviewer‘s comments

We thank the reviewers for their constructive comments. Please find enclosed our responses to the reviewer‘s highlighted in red.

Top of Form

  1. Authors should cut the word “to date” in sentence line no.80, page no. 2 because the research ethics committee of this study has been done since 2012. Amended as suggested (Page no. 2, lines 81-83).
  2.  Authors should explain the detail about a specific study conducted time and the duration study period. Details related to a specific study conducted time and duration period was added as recommended (Page no 2, line 92).
  3. Authors should declare how to calibrate the reliability and internal consistency of the questionnaires.

Cronbach‘s alpha coefficient was calculated to evaluate the internal consistency of the questionnaire and was found to be 0.8 (a good value). Questionnaires‘ validation was peformed through the evaluation of content validity. Finally, validation of questionnaire was performed regarding to the three experts given recommendations. (Page no 3, line 122-129)

  1. How many items are on the questionnaire of adolescent participants? Please specify.

Overall, 20 items comprised the questionnaire for adolescent participants. Background characteristics such as adolescent’s age, gender, grade and living area were analysed. Behavioral factors such as personal oral hygiene skills, measures used for oral hygiene, dental attendance, dietary habits, smoking habits and self-reported oral health status. (Page no 4, lines 130-163)

  1. Authors should discuss demographic characteristics of participants not only maternal and parental education but should also discuss gender and living area.- The main background data such as gender of participants and living area was presented in the Table 1. Gender of adolescent played the role in D-S of active caries lesions among the non-smokers group (Table 5). Further statistical analysis did not show significant differences between living area and other variables, thus it was not presented in the results (Pages no 6, 8)
  2. The one of important factors is socio-economic status but it was not included in this study. Please describe. Considering the negative experience of previously conducted studies, a high percentage of participants tended to skip the question related to household income. Thus, item of household income was not included in this study. Parental education was considered as a social component.
  3. Authors should discuss about the frequency of toothbrushing and consumption of sweets and why were not significant among smoking habits. These factors are important to dental caries and periodontal disease. Authors should discuss these issues. More explanation has now been added as suggested (Pages 2, 10, lines 46-48, 325-326).
  4. Authors should give the details of smoking habit such as the type of cigarette, frequency of smoking per day, and how long the daily smoking habit.

This self-administered questionnaire included two questions related to smoking habit:

 „Have you ever tried to smoke (at least one cigarette)?“ Options of answers: yes or no.

„How often do you smoke at present?“ Option of answers were: ‘daily’, ‘at least once a week, but not daily’, ‘less often than once a week’, or that they ‘do not smoke’.

Participants were not asked about the type of cigarette, smoking duration and the number of smoked cigarettes daily. (Page no 5, lines 138-144)

  1. Authors should explain why the consumption of fruits and vegetables and energy drinks is related to dental caries and periodontal disease.

More explanation has now been added as suggested (pages 2, 10, lines 46-48, 317-318).

  1. Authors should explain why the results of PI were not a significant difference between smoking habit groups but the PSR index showed a significant difference. It might be the other factors that cause periodontal disease. In our opinion, because participants were informed on which oral examination will be performed, they brushed their teeth more srcupulously. Thus PI results did not differ significantly. We thought that current smoking has an impact for PSR index higher score.

  1. Only active caries on proximal has a significant difference between smoking habit groups. So the conclusion from this study might not conclude the smoking group has more tendency to dental caries and periodontal disease than the non-smoking group.

According to the main findings, active caries lesions on proximal and smooth surfaces, and non-cavitated actice caries showed the significant differences between smoking habit groups, as these results were presented in the table 4. In addition, Mean score of PSR index was significantly higher in the currently smoking adolescents‘ group. From our point of view, we might to conclude that current smokers are at higher risk of active dental caries and periodontal diseases (Page 7).

  1. The references need to be replaced with recent references. Approximately 40% of the references should be from the last 2-3 years and 90% from the last 10 years. The list of references was amended as recommended.

Reviewer 3 Report

Dear authors,

Thank you for your submission. We appreciate the time you took to prepare your manuscript. With interest, I have read your manuscript entitled: “A Cross-Sectional Study of Oral Health Status and Behavioral Risk Indicators among Non-Smoking and Currently Smoking Lithuanian Adolescents.” The manuscript is a cross-sectional study that aims to evaluate oral health status, behavioral risk indicators, and the impact of smoking on oral health among Lithuanian adolescents. In general, is a well written paper and the theme of the article is within the journal's editorial line. However, some methodological issues should be revised and evaluated. Some comments below:

Introduction:

1)    The authors stated that “studies investigating associations between oral health and oral-health-related behavioral indicators such as dietary habits, dental attendance, and current levels of tobacco smoking among adolescents has been scarce in Lithuania to date”. However, the lack of studies by itself does not justify another study. What would be the differential of your study in terms of methodology? What does the study advance in the scientific literature? Please revise.

2)    Add the hypothesis of the study.

Methodology:

3)    Please create a separate topic for research ethics committee.

4)    It is not clear if participants signed consent or assent. Consent may only be signed by individuals who have reached the legal age of consent (e.g 18 years old). As the sample is composed of adolescents, the consent of the parent or legal guardian is required and the assent of the adolescent.

5)    Please add a proper reporting guideline for cross-sectional studies.

6)    Was the self-report questionnaire previously validated? Has a test-retest been performed to verify its reliability and internal consistency?

7)    Did the authors perform a pilot study? Were there any changes necessary? Was the sample included in the main study?

8)    It is not clear why only adolescents with 15 years were included.

9)    How was the selection of schools carried out? Were they public? Private? Or both?

Discussion:

10) Once more, the authors stated that this is the first study performed…but, what is the real contribution to scientific literature? And to the population?

Conclusion: Should respond the objectives. Any other information any other information and/or considerations should be written at discussion section.

Author Response

Our responses to the Reviewer‘s comments

We thank the reviewers for their constructive comments. Please find enclosed our responses to the reviewer‘s highlighted in red.

Introduction:

  • The authors stated that “studies investigating associations between oral health and oral-health-related behavioral indicators such as dietary habits, dental attendance, and current levels of tobacco smoking among adolescents has been scarce in Lithuania to date”. However, the lack of studies by itself does not justify another study. What would be the differential of your study in terms of methodology? What does the study advance in the scientific literature? Please revise.

According to our knowledge there were no previously reported studies about oral health status and behavioural risk indicators among non-smoking and currently smoking Lithuanian adolescents. This study focused on active caries lesions, because of possibilities to prevent the further development of disease. We hope that it is new information useful for the scientific society and journal readers.

  • Add the hypothesis of the study.

Hypothesis of the study was added as recommended (Page 2, lines 83-87).

Methodology:

  • Please create a separate topic for research ethics committee.

Ammended as suggested. (Page no 3, lines 114-117)

  • It is not clear if participants signed consent or assent. Consent may only be signed by individuals who have reached the legal age of consent (e.g 18 years old). As the sample is composed of adolescents, the consent of the parent or legal guardian is required and the assent of the adolescent.

Written informed consent for the child‘s oral examination was obtained from the both parents of each adolescent who participated in the study. Permissions were obtained from the schoolchildren themselves as well (Page no 3, lines 100-101).

  • Please add a proper reporting guideline for cross-sectional studies. STROBE checklist was completed as recommended.

  • Was the self-report questionnaire previously validated? Has a test-retest been performed to verify its reliability and internal consistency?

 After 7 days questionnaires were distributed repeatedly for the same respondents to complete. Results of test-retest reliability were equal to 0.77 (0.75-0.8). (Page no 3-4, lines 125-127)

  • Did the authors perform a pilot study? Were there any changes necessary? Was the sample included in the main study?

Pilot study of both clinical examination and questionnaire survey was carried out in the one school („Saule“ gymnasium in the Kaunas City) in January of 2013. Pilot study enrolled thirty-five 15-year-old subjects, who were not included in the main study. Final versions of questionnaires were created after analysis of given comments of both participants and their parents during pilot study (Page no 3, lines 119-121).

  • It is not clear why only adolescents with 15 years were included.

 „According to the WHO, by 15 years, the permanent dentition have been exposed to the oral environment for three to nine years. The age group of 15–19 years is important in the assessment of dental caries and periodontal disease in adolescents.“ This age group is recommended for population surveys by WHO (WHO Oral health surveys- basic methods, 5th edition, 2013).

This research focused on prevalence and experience of dental caries, especially active dental caries lesions, among Lithuanian schoolchildren in late adolescence, therefore 15-year-old and 18-year-old adolescents were enrolled. Articles of data related to 18-year-old schoolchildren were published previously.

  • How was the selection of schools carried out? Were they public? Private? Or both?

This study was performed in the public schools located in the centers of 10 Lithuanian counties. A multistage cluster sampling method was used to define a representative sample. Consequently, each county was divided into smaller administrative units and  schools were picked out from the alphabetic list of all schools, provided by Centre of Information Technologies in Education (the first and the last schools from the list were chosen). (Page 2-3, lines 93-97).

Discussion:

  • Once more, the authors stated that this is the first study performed…but, what is the real contribution to scientific literature? And to the population?

The results of this performed study may be beneficial for the further researches necessity in order to compare the current epidemiological situation. Our study served to create the National Oral Health Project of Lithuania, because this study provided the epidemiological data of therefore 15-year-old and 18-year-old adolescents.

Conclusion:

  • Should respond the objectives. Any other information any other information and/or considerations hould be written at discussion section.

Conclusion was amended as suggested (Page no 11, lines 372-376).

Round 2

Reviewer 3 Report

Dear authors, the manuscript had improved after following the valuable comments of the reviewers. The paper is now adequate.